# SpaceControl: Introducing Test-Time Spatial Control to 3D Generative Modeling

**Elisabetta Fedele**[1,2*], **Francis Engelmann**[2,3*], **Ian Huang**[2], **Or Litany**[4,5],
**Marc Pollefeys**[1], **Leonidas Guibas**[2]
[1]ETH Zurich   [2]Stanford University   [3]USI Lugano   [4]Technion   [5]NVIDIA   *Equal contribution

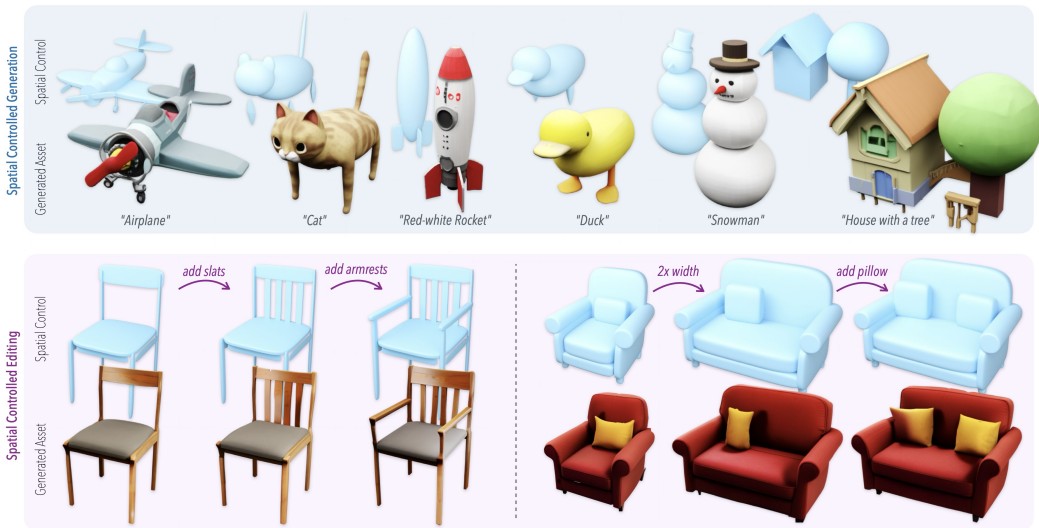

**Figure 1:** SpaceControl enables spatially controlled 3D asset generation from simple geometric primitives such as *superquadrics (light blue)* and other geometry types such as polygon meshes. **Top:** Rapid asset generation. From quick 3D sketches and brief text prompts, we can generate high quality assets. **Bottom:** Fine-grained editing, including adjusting a chair's backrest and adding armrests *(left)* or precisely controlling a sofa's dimensions and pillow arrangements *(right)*.

## Abstract

Generative methods for 3D assets have recently achieved remarkable progress, yet providing intuitive and precise control over the object geometry remains a key challenge. Existing approaches predominantly rely on text or image prompts, which often fall short in geometric specificity: language can be ambiguous, and images are difficult to manipulate. In this work, we introduce SpaceControl, a training-free test-time method for explicit spatial control of 3D asset generation. Our approach accepts a wide range of geometric inputs, from coarse primitives to detailed meshes, and integrates seamlessly with modern generative models without requiring any additional training. A control parameter lets users trade off between geometric fidelity and output realism. Extensive quantitative evaluation and user studies demonstrate that SpaceControl outperforms both training-based and optimization-based baselines in geometric faithfulness while preserving high visual quality. Finally, we present an interactive interface for real-time superquadric editing and direct 3D asset generation, enabling seamless use in creative workflows. Project page: https://spacecontrol3d.github.io/.

## 1 Introduction

Generating 3D assets is a fundamental step in building virtual worlds, useful for gaming, simulation, virtual reality applications, and digital design. Recently the field of 3D object generation gained immense traction, and we are now able to create assets of previously unseen quality (Xiang et al., 2025; Zhang et al., 2024; Vahdat et al., 2022; Gao et al., 2022; Wu et al., 2025; Siddiqui et al., 2024;

Zhao et al., 2025; Chen et al., 2025b; Huang et al., 2025; Deb Sarkar et al., 2025). A continuing difficulty, however, is *controllability*, *i.e.*, the ability for users to reliably guide the object generation process so that it matches the intended shapes and geometric characteristics.

Most controllable 3D generation approaches are conditioned on either *text* or *images*. Text is flexible and easy to use, but its ambiguity makes it poorly suited for specifying precise geometry. Images better constrain 3D structure, yet they are difficult to edit and unintuitive for fine-grained control. Consequently, neither modality allows artists or designers to directly manipulate the object geometry. The aim of this work is to lower the barrier to 3D asset creation by shifting control into *3D space* itself. Rather than relying on text or image conditioning, this work introduces what we call *spatial control*: the use of 3D geometry as three-dimensional sketches that steer the synthesis of detailed 3D assets.

Existing approaches that provide spatially controlled 3D object generation fall into two categories: *training-based* and *guidance-based* methods. The former extend generative models by fine-tuning them to accept particular forms of geometric inputs, for example, voxel conditioning in LION (Vahdat et al., 2022) or primitive- and mesh-based inputs in Spice-E (Sella et al., 2024). Such approaches deliver explicit spatial control while maintaining the inference time of the base model, but require additional training, which often reduces their generalization capabilities. By contrast, guidance-based methods such as LatentNeRF (Metzer et al., 2023) operate solely at inference time, avoiding retraining, but typically incur substantial optimization overhead. Other approaches combine fine-tuning with test-time optimization. For instance, Coin3D (Dong et al., 2024) fine-tunes a multi-view generative model to produce geometry-consistent images, which are then used jointly with the geometric conditioning to optimize a geometry-aware radiance field.

Another line of works focuses on enriching existing 3D assets with geometric and appearance details (Michel et al., 2022; Chen et al., 2023; Barda et al., 2025), but typically assumes fine-grained input geometry, limiting its usefulness in creative workflows where artists tend to begin with coarse sketches.

In this work, we present SPACECONTROL, a training-free method that introduces explicit spatial control into modern generative frameworks for text- or image-conditioned 3D generation, such as Trellis (Xiang et al., 2025) or SAM 3D (Chen et al., 2025a). The approach directly encodes user-specified geometry into the latent space as explicit guidance, requiring no additional training and enabling controllable generation from a broad range of geometric inputs, from simple geometric primitives to detailed meshes.

We evaluate SPACECONTROL against existing training-based (Sella et al., 2024) and guidance-based (Dong et al., 2024) approaches, as well as a stronger training-based variant of Spice-E adapted for Trellis. Despite requiring no fine-tuning, SPACECONTROL achieves superior geometric faithfulness while maintaining high visual realism. Furthermore, we introduce a user interface for interactive superquadric editing and real-time textured asset generation, facilitating seamless integration into practical design flows.

In summary, our contributions are as follows:

- We introduce a training-free guidance method that conditions a powerful pre-trained generative model (Xiang et al., 2025) on user-defined geometry via latent-space intervention, enabling geometry-aware generation without costly fine-tuning.
- We provide extensive evaluations, including quantitative analysis and a user study, demonstrating that our method outperforms prior state-of-the-art approaches for shape-conditioned 3D asset generation.
- We develop an interactive interface for intuitive superquadric editing and real-time conversion into detailed, textured 3D assets, facilitating practical use in creative workflows.

## 2 RELATED WORK

### 2.1 3D GENERATIVE MODELS

The field of 3D generation has experienced a rapid growth during the past few years both in terms of output modalities and controllability. Similar to the first image diffusion models (Ramesh et al.,

2021), early applications of diffusion models for 3D generation (Nichol et al., 2022) were conducting the diffusion process in the original input space and were limited in the generated output type. More recent approaches (Vahdat et al., 2022; Jun & Nichol, 2023) started running the generation in a more compact latent space, leading to substantial improvements both in terms of quality and efficiency. To achieve greater efficiency, Zhang et al. (2024); Xiang et al. (2025) have begun to disentangle geometric structure from its appearance, leading to unprecedented quality in 3D generations. This separation enables the isolated modification of geometry via *spatial* conditioning, which we introduce in this work.

## 2.2 CONTROLLABLE GENERATIVE MODELS

Given a pretrained generative model, there are two main approaches to introduce a new control modality: (1) methods which *finetune* a part or the whole network to take new types of conditioning as input, and (2) *training-free* methods which condition the generation via inference-time guidance. In the last years, many approaches have been developed to control image generative models (Meng et al., 2022; Zhang et al., 2023), enabling conditioning in several forms such as strokes or depth maps. In contrast, analogous mechanisms for 3D object generation remain far less mature.

### CONTROLLING IMAGE GENERATIVE MODELS

A wide variety of methods have been proposed to introduce new control modalities to image generative models. Among works based on fine-tuning, we identify two main lines of research. One category follows ControlNet (Zhang et al., 2023; Bhat et al., 2024), which adds conditional control by introducing a trainable copy of the network connected via zero convolutions. The key idea is to learn control without discarding information from the original pre-trained model. Alternatively, other approaches add specialized layers to provide additional network control (Garibi et al., 2025; Hertz et al., 2022). Among training-free methods (Von Rütte et al., 2024; Meng et al., 2022; Sajnani et al., 2025), SDEdit (Meng et al., 2022) is closely related to our work. It leverages the denoising process of SDE-based generative models by using for example stroke paintings to initialize and condition image generation.

### CONTROLLING 3D GENERATIVE MODELS

Only a few works have explored spatially grounded control of 3D generative models. One category of approaches, including Latent-NeRF (Metzer et al., 2023), Coin3D (Dong et al., 2024), Fantasia3D (Chen et al., 2023), Instant3dit (Barda et al., 2025), and Phidias (Wang et al., 2025), applies spatial control to image generative models by projecting the 3D conditioning onto multiple views and leveraging test-time optimization to synthesize a 3D representation. Alternatively, Spice-E (Sella et al., 2024) applies control directly in 3D space by fine-tuning Shap-E (Jun & Nichol, 2023) on specific ShapeNet (Chang et al., 2015) categories. While both strategies attempt explicit spatial control, they fall short of the flexibility seen in 2D counterparts. The former requires lengthy optimization and conditions 2D projections rather than the 3D volume directly. The latter necessitates class-specific fine-tuning, which limits generalization and prevents tuning the strength of the geometric control. In contrast, SPACECONTROL introduces test-time guidance directly within 3D generative models, providing a framework that is efficient, accurate, and fast.

## 3 PRELIMINARIES

Before introducing our SPACECONTROL, we review the foundations on which it builds: rectified flow matching, the Trellis (Xiang et al., 2025) generative model, as well as superquadrics.

### 3.1 RECTIFIED FLOW MODELS

Rectified flow models use a linear interpolation forward (diffusion) process where for a specific time step $t \in [0, 1]$, the latent $\mathbf{z}_t$ can be expressed as $\mathbf{z}_t = (1 - t)\mathbf{z}_0 + t\epsilon$, where $\epsilon \sim \mathcal{N}(\mathbf{0}, I)$ and $\mathbf{z}_0$ is a clean sample from the target data distribution. The backward (denoising) process is represented by a time dependent velocity field $\mathbf{v}(\mathbf{z}_t, t) = \nabla_t \mathbf{z}_t$. In practice, starting from a noisy sample $\mathbf{z}_1$, we can obtain the denoised version $\mathbf{z}_0$ by discretizing the time interval $[0, 1]$ into $T$ discrete steps, possibly

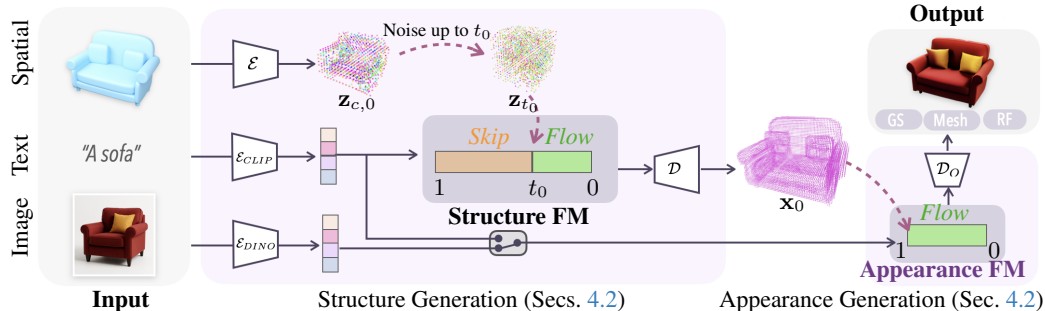

**Figure 2: Model Overview.** Given an input conditioning which includes a spatial control, a text prompt and an image (optional), SPACECONTROL produces realistic 3D assets. First the different conditioning are encoded in a latent space. Specifically, the spatial control is voxelized and encoded by Trellis' encoder $\mathcal{E}$, the text is encoded by a CLIP encoder $\mathcal{E}_{CLIP}$, and the image (if present) is encoded by a DINOv2 encoder $\mathcal{E}_{DINO}$. The obtained latents $\mathbf{z}_{0,c}$ are noised up to $t_0$ to obtain $\mathbf{z}_{t_0}$. From $t_0$ to $t = 0$, $\mathbf{z}_{t_0}$ are denoised by the *Structure Flow Model* (FM), guided by the text prompt features. The clean latents $\mathbf{z}_0$ are then fed into the decoder $\mathcal{D}$, which outputs the voxel grid $\mathbf{x}_0$. Then, the active voxels are augmented with point-wise noisy latent features, denoised by the *Appearance Flow Model* (FM), using either text or image conditioning. The clean latents can then be decoded into versatile output formats such as 3D gaussians (GS), radiance fields (RF), and meshes (M) via specific decoders $\mathcal{D}_O = \{\mathcal{D}_{GS}, \mathcal{D}_{RF}, \mathcal{D}_M\}$.

not uniformly distributed, and recursively applying the equation

$$\mathbf{z}_{t(i+1)} = \mathbf{z}_{t(i)} - \mathbf{v}_\theta\big(\mathbf{z}_{t(i)}, t(i)\big)\big(t(i) - t(i+1)\big),\tag{1}$$

where $i \in [1, T-1]$, and the vector field $\mathbf{v}_\theta(\cdot)$ is predicted for example by a Diffusion Transformer (Peebles & Xie, 2023) as in Trellis (Xiang et al., 2025) or SAM 3D (Chen et al., 2025a).

**Step Scheduler.** Time steps are initially defined as $t(\tau) = 1 - \tau/T$ for $\tau \in [0, T]$, and then rescaled by a factor $\lambda$:

$$t(\tau) = \frac{\lambda t(\tau)}{1 + (\lambda - 1)t(\tau)}.\tag{2}$$

Since $t$ can be obtained from $\tau$ and vice versa, we will refer to either one interchangeably.

## 3.2 TRELLIS

Trellis (Xiang et al., 2025) is a recent 3D generative model which employs rectified flow models to generate 3D assets from either textual or image conditioning. Specifically, it consists of two separate steps of generations, where the first aims to generate the geometric *structure*, while the second focus on the *appearance*. Note that SPACECONTROL is equally applicable to more recent works such as SAM 3D (Chen et al., 2025a), which follow the same two-step generation approach.

**Structure Generation.** In the first stage, a noisy latent variable $\mathbf{z}_1 \in \mathbb{R}^{16 \times 16 \times 16 \times 8}$ is sampled from $\mathcal{N}(\mathbf{0}, I)$ and denoised by a rectified flow model iteratively applying Eq. 1 using either image or text conditioning. Specifically, text conditions are encoded via the CLIP (Radford et al., 2021) text encoder, while image conditions are encoded via DINOv2 (Oquab et al., 2024). The denoised latent $\mathbf{z}_0$ is then decoded by a decoder $\mathcal{D}$ into a binary occupancy grid $\mathbf{x} \in \{0, 1\}^{64 \times 64 \times 64}$, which represents the coarse spatial structure of the generated 3D asset. Note that the pretrained decoder $\mathcal{D}$ is accompanied by a corresponding pretrained encoder $\mathcal{E}$. While $\mathcal{E}$ is not utilized during standard Trellis inference, it is essential for our guidance mechanism, as it maps the spatial conditioning signal into the shared latent space (see Sec. 4).

**Appearance Generation.** In the second stage, the $L$ active voxels of the binary occupancy grid are augmented with point-wise noisy latent features $\mathbf{s}_1 \in \mathbb{R}^{L \times 8}$ sampled from $\mathcal{N}(\mathbf{0}, \mathbf{I})$, which are denoised by a second flow model using either text or image conditioning. The clean latents $\mathbf{s}_0 \in \mathbb{R}^{L \times 8}$ can then be decoded into versatile formats such as 3D Gaussians (GS), Radiance Fields (RF), and Meshes (M) via specific decoders $\mathcal{D}_O = \{\mathcal{D}_{GS}, \mathcal{D}_{RF}, \mathcal{D}_M\}$. We refer the reader to Xiang et al. (2025) for further details.

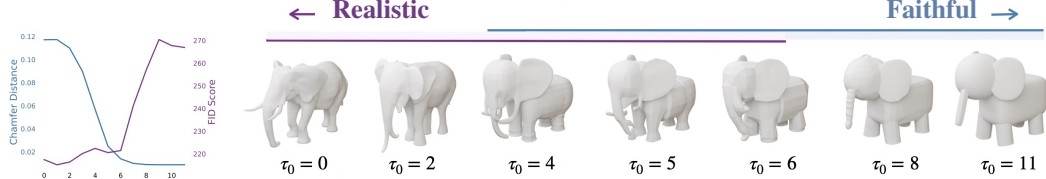

**Figure 4: Realism-faithfulness tradeoff.** The hyperparameter $\tau_0$ allows a smooth control over the strength of the control. In the left figure we show how variations of $\tau_0$ affects the generations quantitatively in terms of Chamfer distance to the spatial control (lower means more *faithful*) and of FID score (lower means more *realistic*). In the right figure we show it qualitatively, visualizing how higher values of $\tau_0$ lead to assets whose geometry looks even more similar to the control. For conciseness we only show the untextured geometry.

**Superquadrics.** Superquadrics (Barr, 1981) provide a compact parametric family of shapes capable of representing diverse geometries. A canonical superquadric is defined by five parameters: scales $(s_x, s_y, s_z)$ and exponents $(\epsilon_1, \epsilon_2)$. With parametric coordinates $(\eta, \omega)$ we can define their surface as:

$$s(\eta, \omega) = \begin{bmatrix} s_x \cos(\eta)^{\epsilon_1} \cos(\omega)^{\epsilon_2} \\ s_y \cos(\eta)^{\epsilon_1} \sin(\omega)^{\epsilon_2} \\ s_z \sin(\eta)^{\epsilon_1} \end{bmatrix}. \tag{3}$$

Extending to world coordinates requires 6 additional pose parameters (3 translation, 3 rotation), giving 11 parameters in total. Their compactness makes them well-suited as spatial control primitives.

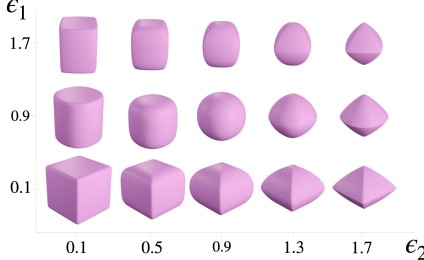

**Figure 3:** Superquadric primitives.

# 4 METHOD

## 4.1 SETUP

To introduce spatial control in 3D generation, the user provides a geometric conditioning signal alongside a text or image prompt. Our goal is to produce 3D assets that satisfy two key criteria:

- **Faithfulness**: the generated asset should be aligned with the control geometry.
- **Realism**: the generated asset should retain the quality of the original model.

## 4.2 APPROACH

Next, we introduce SPACECONTROL (see Fig. 2) and describe how it enables guided 3D asset generation by injecting spatial guidance into a pretrained Trellis model. Because our control strategy differs between the first and second stages of generation, we outline each procedure in Secs. 4.2 and 4.2, respectively.

**Structure generation.** To control the first step of generation given an explicit control geometry we employ a similar framework to SEdit (Meng et al., 2022), where instead of using *strokes* to guide the generation of *2D images*, we use either coarse or detailed 3D geometry to guide the generation of *3D assets*. Specifically, given a user-specified 3D geometry, we voxelize it to obtain $\mathbf{x}_c \in \{0, 1\}^{64 \times 64 \times 64}$ and feed $\mathbf{x}_c$ into the pretrained encoder $\mathcal{E}$ to obtain $\mathbf{z}_{c,0} \in \mathbb{R}^{16 \times 16 \times 16 \times 8}$. Then given a specific time step $t_0 \in [0, 1]$ we noise up the latents $\mathbf{z}_{c,0}$ to that specific step via the rectified flows forward equation:

$$\mathbf{z}_{t_0} = t_0 \mathbf{z}_1 + (1 - t_0) \mathbf{z}_{c,0}, \tag{4}$$

where $\mathbf{z}_1 \sim \mathcal{N}(\mathbf{0}, I)$. Given $\mathbf{z}_{t_0}$, $\mathbf{z}_0$ can then be obtained by iteratively applying Eq. 1 starting from $t_0$ and employing the original *Structure Flow Model*. We note that this process does not require any need of architectural changes nor training. We guide the generation with additional textual prompt, which is helpful to disambiguate the semantics of the object. As in the standard setting, the denoised latent $\mathbf{z}_0$ is then decoded into a final geometric structure $\mathbf{x}_0 \in \{0, 1\}^{64 \times 64 \times 64}$ by $\mathcal{D}$.

**Appearance generation.** Given the geometric structure generated in the first stage, we then employ either text or image conditioning to guide the generation of its appearance, by first expanding the active voxels with point-wise noisy latent features and then denoising them using the *Appearance Flow Model*. Notice that, even if the structure generation is always conditioned on text, image conditioning can still be used to guide the appearance generation, allowing for finer control over the visual details (see Fig. 7a and Appendix).

**Controlling the strength of spatial control.** The strength of spatial control can be tuned through the parameter $\tau_0$. For lower values of $\tau_0$, the latent $z_{t_0}$ is initialized closer to the noise $z_1$ than to the control signal $z_{c,0}$, leading the model to perform more denoising steps. This favors samples that follow the data distribution of the original Trellis, producing outputs that are generally more realistic but less faithful to the spatial conditioning. In contrast, higher values of $\tau_0$ bias $z_{t_0}$ towards $z_{c,0}$, effectively skipping earlier denoising steps and preserving more of the injected spatial structure, albeit sometimes at the expense of realism.

## 5 EXPERIMENTS

### 5.1 COMPARING WITH STATE-OF-THE-ART METHODS

**Tasks.** We evaluate SPACECONTROL under two types of spatial conditions: (1) coarse geometry and (2) detailed geometry, using simple geometric primitives and full object meshes, respectively.

**Baselines.** We compare SPACECONTROL to state-of-the-art baselines for the task of 3D-conditioned object generation. We evaluate against Spice-E (Sella et al., 2024), which fine-tunes Shap-E (Jun & Nichol, 2023) to support cuboid primitives as spatial guidance. To ensure a fairer comparison between backbones, we implement a corresponding version for Trellis (Xiang et al., 2025), which we refer to as SPICE-E-T; details on its implementation and training are provided in

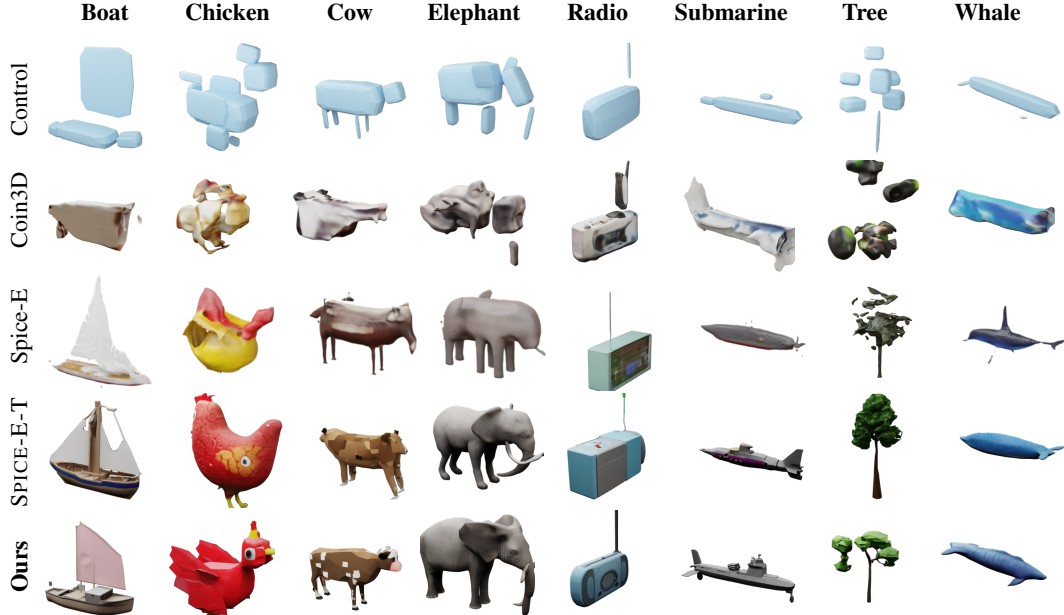

Figure 5: **Qualitative Comparison of Spatially Conditioned Generation.** We show generations obtained conditioning our SPACECONTROL and baselines on text prompts and superquadrics from the Toys4K dataset. While other methods either fail to follow the conditioning (*e.g.*, the antenna from the radio generated by Spice-E is wrongly placed) or to generate visually appealing 3D assets (*e.g.*, the chicken generated by SPICE-E-T exhibits anatomically incorrect body part placements), SPACECONTROL exhibits a good balance between realism and faithfulness.

**Table 1: Comparison with Baselines.** We evaluate faithfulness to spatial and textual control via Chamfer Distance (CD, $\times 10^3$) and CLIP-I, and realism via FID (texture) and P-FID (geometry). Results for SPACE-CONTROL use $\tau_0 = 6$. [†] indicates methods fine-tuned on *chair* and *table* categories. Trellis (Xiang et al., 2025) (txt-DiT-XL) is included for reference only as it does not support spatial guidance.

| Method | Toys4K | | | | Chair | | | | Table | | | |
|---|---|---|---|---|---|---|---|---|---|---|---|---|
| | CD↓ | CLIP-I↑ | FID↓ | P-FID↓ | CD↓ | CLIP-I↑ | FID↓ | P-FID↓ | CD↓ | CLIP-I↑ | FID↓ | P-FID↓ |
| Trellis | 117 | 0.33 | 217 | 78.60 | 14.7 | 0.31 | 129 | 40.82 | 19.7 | 0.30 | 132 | 49.40 |
| **Geometric Primitives** | | | | | | | | | | | | |
| Coin3D | 54.4 | 0.21 | 231 | 102.0 | 18.5 | 0.25 | 218 | 47.54 | 28.82 | 0.22 | 245 | 71.58 |
| Spice-E[†] | 65.9 | 0.29 | 233 | 66.52 | 7.66 | 0.29 | 166 | 38.66 | 10.3 | 0.29 | 148 | 78.85 |
| SPICE-E-T[†] | 39.1 | **0.32** | 223 | **53.51** | 5.92 | **0.31** | 135 | 39.22 | 4.73 | **0.30** | **122** | 47.36 |
| SPACECONTROL (Ours) | **14.0** | **0.32** | 221 | 81.3 | **0.98** | 0.30 | 146 | **34.06** | **3.72** | 0.29 | 157 | **46.28** |
| **Meshes** | | | | | | | | | | | | |
| Coin3D | 77.8 | 0.04 | 293 | 182.5 | 14.6 | 0.01 | 308 | 111.0 | 20.4 | 0.01 | 224 | 178.2 |
| Spice-E (stylization) | 7.40 | 0.30 | 224 | 81.21 | 6.37 | 0.30 | 152 | 41.51 | 28.2 | 0.29 | 132 | 58.01 |
| SPICE-E-T[†] | 23.3 | **0.32** | **222** | 90.99 | 22.7 | **0.31** | **132** | 39.70 | 7.59 | **0.30** | **116** | 46.76 |
| SPACECONTROL (Ours) | **4.89** | 0.29 | 244 | **72.47** | **0.66** | 0.29 | 137 | **30.96** | **0.48** | 0.28 | 130 | **42.33** |

the Appendix. We note that Spice-E utilizes a separate checkpoint for shape stylization when evaluating mesh conditioning, as it yields superior results. Additionally, we compare to Coin3D (Dong et al., 2024), which leverages shape guidance to first generate a single view of the desired 3D asset, then uses a fine-tuned multi-view diffusion model to generate consistent views, and finally extracts the 3D representation via volumetric-based score distillation sampling.

**Datasets.** To evaluate how various approaches handle geometric conditioning, we construct a dataset comprising original meshes, their respective geometric primitive decompositions, and corresponding textual descriptions. This dataset allows for a two-fold evaluation: assessing methods on mesh-conditioned generation using the original assets and on shape-conditioned generation via the primitives. To measure both generation and generalization performance, our evaluation spans two ShapeNet (Chang et al., 2015) categories (chairs and tables) included in the Spice-E training set, alongside objects from the Toys4K (Stojanov et al., 2021) dataset, which remain unseen by all methods during training. We utilize SuperDec (Fedele et al., 2025) to decompose the 3D assets into superquadrics and employ Gemini on rendered views to generate textual descriptions for the ShapeNet assets. For the Toys4K objects, we adopt the textual descriptions provided by Trellis.

**Metrics.** Our experiments aim to evaluate both the *faithfulness* to the spatial and textual control and the *realism* of the generated assets. Faithfulness to the spatial control is quantified using the L2 *Chamfer Distance* (CD) between vertices sampled from the input superquadric primitives and the generated mesh decoded by $\mathcal{D}_M$. Faithfulness to the textual control is quantified with the CLIP similarity (CLIP-I) between the renderings of generated assets and the textual prompts. Realism is evaluated for texture via the *Fréchet Inception Distance* (FID) (Heusel et al., 2017) on image renderings and for geometry, via the P-FID (Nichol et al., 2022), the point cloud analog for FID. To measure the FID on image rendering we measure the distance between the inception features extracted from the original image renderings of the datasets and the generated ones. To measure the P-FID of the generated meshes we measure the distance between the PointNet++ (Qi et al., 2017) features of the generated and original object meshes.

**Results.** Quantitative results are reported in Tab. 1, while qualitative results are shown in Fig. 5. Both Spice-E and SPICE-E-T perform well on *chairs* and *tables* but struggle to faithfully generate objects that they were not fine-tuned on (Toys4K). SPACECONTROL significantly outperforms the baselines in all experiments in terms of Chamfer Distance (CD) to the spatial control, while achieving comparable CLIP-I, FID, and P-FID scores. For completeness, we also report scores for the text-conditioned Trellis using the DiT-XL backbone, which is also the base model used in our SPACECONTROL. Note that for the sake of simplicity in Tab. 1 we only report results of SPACE-CONTROL with $\tau_0 = 6$. However, $\tau_0$ can be chosen freely by the user, depending on the desired strength of conditioning. For completeness, we report results for different values of $\tau_0$ in Tab. 2. We

**Table 2: Analysis of $\tau_0$.** We evaluate faithfulness to spatial and textual control via Chamfer Distance (CD, scaled by $10^3$) and CLIP-I, respectively. Realism is assessed via FID for texture and P-FID for geometry. Results are reported for spatial control provided as geometric primitives (**P**) and meshes (**M**).

| | Toys4K | | | | | | | | Chair | | | | | | | | Table | | | | | | | |
|---|---|---|---|---|---|---|---|---|---|---|---|---|---|---|---|---|---|---|---|---|---|---|---|---|
| | CD↓ | | CLIP-I↑ | | FID↓ | | P-FID↓ | | CD↓ | | CLIP-I↑ | | FID↓ | | P-FID↓ | | CD↓ | | CLIP-I↑ | | FID↓ | | P-FID↓ | |
| $\tau_0$ | P | M | P | M | P | M | P | M | P | M | P | M | P | M | P | M | P | M | P | M | P | M | P | M |
| 0 | 117 | 75.4 | **0.33** | **0.29** | 217 | 254.9 | **78.6** | 79.4 | 14.7 | 30.6 | **0.31** | **0.29** | 129 | 133.7 | 40.8 | 39.9 | 19.7 | 49.21 | **0.30** | **0.28** | 132 | 137.5 | 49.40 | 49.3 |
| 2 | 110 | 65.5 | **0.33** | **0.29** | 216 | 256.9 | 79.1 | 82.7 | 14.1 | 30.0 | **0.31** | **0.29** | 131 | 136.7 | 41.2 | 41.5 | 18.5 | 43.51 | **0.30** | **0.28** | 132 | 134.7 | 51.97 | **41.5** |
| 4 | 56.8 | 32.4 | 0.32 | **0.29** | 222 | 252.8 | 84.1 | 83.9 | 7.3 | 13.9 | **0.31** | **0.29** | 137 | 141.1 | 34.1 | 31.9 | 6.33 | 2.68 | **0.30** | **0.28** | 135 | 133.5 | 51.79 | 45.8 |
| 6 | 14.0 | 4.89 | 0.32 | **0.29** | 221 | 244.9 | 81.3 | **72.5** | 0.98 | 0.66 | 0.30 | **0.29** | 146 | 136.6 | **34.0** | 31.0 | 3.72 | 0.48 | 0.29 | **0.28** | 157 | 131.0 | **46.28** | 42.3 |
| 8 | 9.04 | 1.57 | 0.29 | **0.29** | 257 | 241.3 | 94.0 | 77.0 | 0.27 | 0.28 | 0.30 | 0.28 | 156 | 134.3 | 37.1 | **29.2** | 3.29 | 0.19 | 0.29 | **0.28** | 175 | 127.3 | 50.16 | 43.2 |
| 10 | **8.85** | **1.84** | 0.27 | **0.29** | 268 | **209.3** | 101 | 74.9 | **0.22** | **0.26** | 0.30 | 0.28 | 160 | 134.0 | 36.5 | 30.1 | **3.26** | 0.19 | 0.29 | 0.29 | 181 | **125.9** | 50.74 | 42.6 |

can see that by increasing the value of $\tau_0$ and thus strength of the spatial conditioning, we obtain generations which align more closely to the input spatial control.

**User Study.** To validate the numerical results, we conduct a user study (Fig. 6) involving 52 volunteers, each one evaluating on average 20 randomly selected samples. Participants were asked to compare pairs of generated objects, voting which one was more faithful to the input control shape, which model looked more realistic, and which one they liked overall better (see Appendix for more details). The study is performed on the same datasets discussed above, *i.e.* on ShapeNet (Chang et al., 2015) and Toys4k (Stojanov et al., 2021). We compare our SPACECONTROL to the Spice-E and Spice-E-T baselines. We observe that our SPACECON-TROL is always the preferred method both in terms of overall appearance and alignment to the input spatial control.

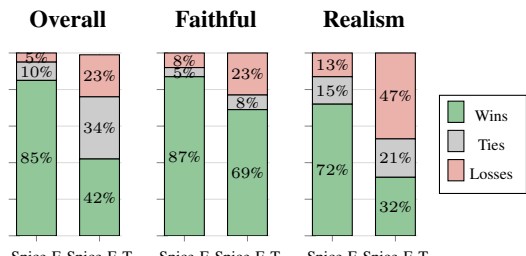

**Figure 6: User Study Results.** The bar plots present the proportion of favorable comparisons achieved by our SPACECONTROL against the baselines on overall appearance, faithfulness to spatial control, and realism, respectively.

**Qualitative results** are shown in Fig. 1, 5, and 7. Additional results for object editing are in the Appendix, visualizing outputs of different methods conditioned on both coarse and detailed input controls. In general, training-based methods struggle to generate objects in specific poses, whereas SPACECONTROL consistently produces plausible results. For example, other methods generate a cow with two heads (Spice-E and Spice-E-T), an elephant with an eye on its back (Spice-E), or shapes that fail to strictly follow the spatial conditioning or exhibit low quality (Coin3D).

## 5.2 Analysis experiments

**The Effect of the Control Parameter $\tau_0$.** While existing methods for 3D spatial conditioning do not provide a way to control its strength, our SPACECONTROL enables flexible interpolation between different levels of adherence. In this section, we evaluate how the parameter $\tau_0$ governs the trade-off between fidelity to the spatial control signal and the realism of the generated asset. Quantitative results are reported in Tab. 2, using the same metrics and datasets as in Tab. 1. We further present qualitative results in Fig. 4 and in the Appendix, showing how varying the conditioning strength produces different outcomes. Adjusting $\tau_0$ allows users to regulate this trade-off according to their preferences, balancing higher shape quality against stronger adherence to the spatial guidance. Additionally, the plot in Fig. 4 *(left)* illustrates this trade-off on Toys4K, indicating that $\tau_0 \in [4, 6]$ generally provides a good compromise between spatial adherence and shape quality.

**The Role of Image Conditioning.** SPACECONTROL supports multi-modal control for 3D asset generation by combining spatial guidance with natural language and optional image conditioning. While the model can synthesize assets using geometric conditioning and textual prompts alone, images are particularly useful for maintaining visual consistency during object edits, as shown in Fig. 7a and in the Appendix. Moreover, as we only use image prompts in the *Appearance Flow*

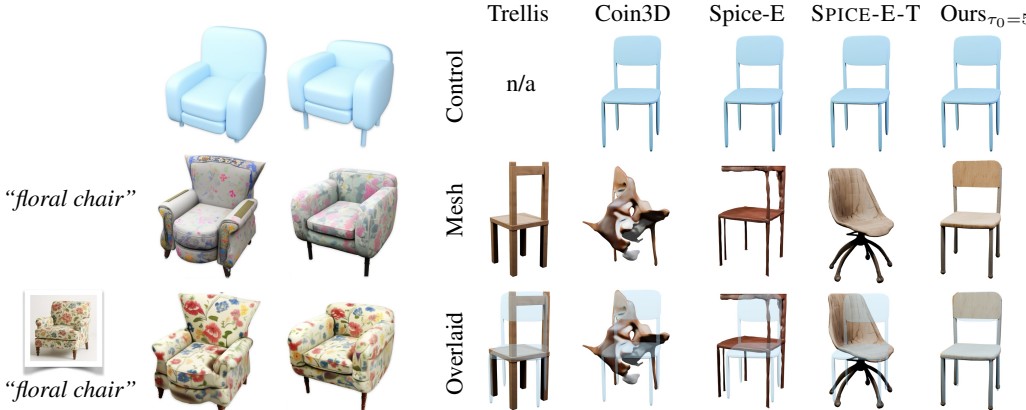

(a) **Image Conditioning.** Given the two different spatial controls shown in the *first row*, we show objects generated by our SPACECONTROL without *(second row)* and with *(third row)* image conditioning.

(b) **Spatial alignment.** We show how different methods align the generated 3D asset with the input condition. In the first row we show the input control, in the second the generated asset and in the third, we overlay the two. All the generations use the same prompt *"A wooden chair."*.

**Figure 7: Image conditioning and fine-grained alignment.** We show analysis experiments on the role of image conditioning (*left*) and on fine-grained spatial alignment (*right*).

*Model* of Trellis, they primarily affect texture, with only minor influence on the geometry. While this capability originates from the pre-trained Trellis, SPACECONTROL enables its practical use for cross-modal texture transfer, effectively performing style transfer from 2D images to generated 3D shapes.

**Spatial Alignment.** We believe that a key advantage of a training-free approach that performs conditioning directly in 3D space is its ability to achieve fine-grained spatial control. In this section, we provide an example where the conditioning shapes are not aligned with axis-oriented rotations. As shown in Fig. 7b, our method is the only one that perfectly aligns with the input conditioning while preserving the quality of the generated mesh. Additional results are provided in the Appendix.

## 6 DISCUSSION AND CONCLUSION

In summary, SPACECONTROL introduces the first training-free framework that operates directly in 3D space to enable precise spatial conditioning for high-quality asset generation.Future Work and Potential. The flexibility of SPACECONTROL provides several promising directions for future exploration. Currently, the adherence parameter $\tau_0$ allows users to manually navigate the realism–faithfulness tradeoff. While this enables fine-grained user control, future work could investigate adaptive or learned scheduling for $\tau_0$ based on class-specific geometry or sample density. Furthermore, our formulation establishes a foundation for part-aware control; extending the global adherence mechanism to localized regions would allow for varying degrees of geometric fidelity within a single object.Beyond individual assets, our method's training-free nature makes it naturally extensible to complex 3D scenes. The integration of scene abstraction techniques like SuperDec (Fedele et al., 2025) or Search3D (Takmaz et al., 2025) offers a path toward generating realistic spatial layouts from geometric primitives. Moreover, incorporating functional scene graphs (Zhang et al., 2025) would allow SPACECONTROL to respect semantic relationships between objects, ultimately enabling the automated generation of "digital cousins" (Dai et al., 2024) for real-world environments.

**Reproducibility Statement.** Our approach builds on the open-source Trellis model (Xiang et al., 2025), and our experiments use open-source datasets, namely ShapeNet (Chang et al., 2015) and Toys4k (Stojanov et al., 2021). Our code is publicly available on our project page.

**Acknowledgments.** Elisabetta Fedele acknowledges support from the ETH AI Center doctoral fellowship, the Swiss National Science Foundation (SNSF) Advanced Grant 216260 (*Beyond Frozen Worlds: Capturing Functional 3D Digital Twins from the Real World*), and an SNSF Mobility Grant. Francis Engelmann acknowledges support from an SNSF PostDoc mobility fellowship. Or Litany acknowledges support from the Israel Science Foundation (grant 624/25) and the Azrieli Foundation Early Career Faculty Fellowship. This research was also supported in part by an academic gift from NVIDIA and Meta. The authors also acknowledge the support from a SwissAI Grant for Small Projects and an Academic Grant from NVIDIA.

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

# A    ADDITIONAL RESULTS

## A.1    FINE-GRAINED SPATIAL EDITING

In this section we provide additional results which show how the generations from our SPACECON-TROL are influenced by the change of the spatial control. We show results in pairs where the textual and/or image prompts are kept fixed. We notice that by providing additional image control, we are able to preserve the texture between different generations.

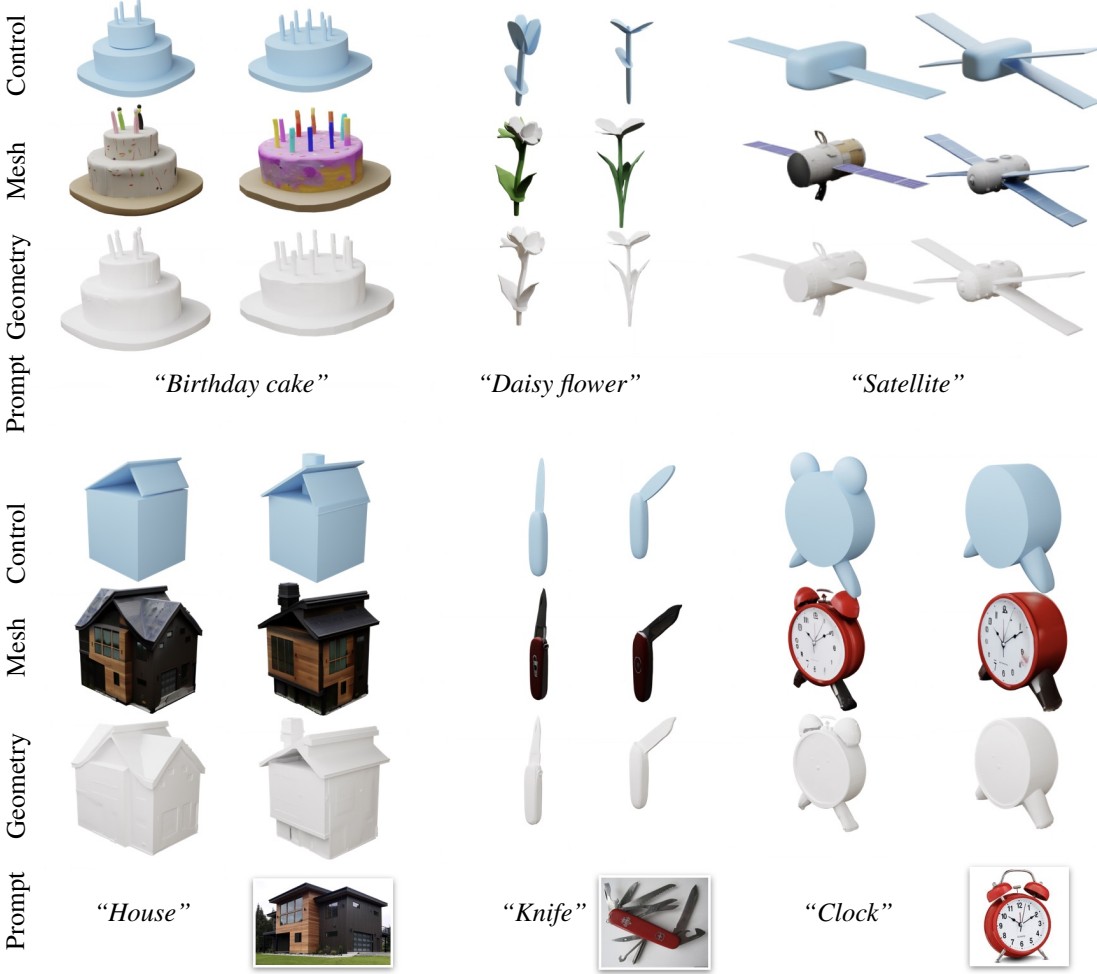

Figure 8: **Fine-grained spatial editing with superquadrics.** Superquadrics offer fine-grained spatial control that is useful not only for generating a wide variety of 3D assets, but also for editing them. They enable intuitive and localized modifications of 3D shapes, in a more direct manner than text- or image-only generative models in practicality. In addition to natural language prompts (*top*), SPACECONTROL supports image conditioned generation (*bottom*), enabling consistent visual appearance across edits.

## A.2 COARSE AND FINE-GRAINED SPATIAL CONTROL WITH SUPERQUADRICS

In this section, we provide additional results generated with different control strengths. Here the hyperparameter is chosen so that we were satisfied with the final result. Superquadrics prove to be an effective tool to provide both coarse and fine-grained control to the 3D generation. By combining the expressivity of superquadrics with the flexible control strength offered by our SPACECONTROL, users can condition the generation by either carefully designing geometric details or only drafting the spatial setting of the desired output.

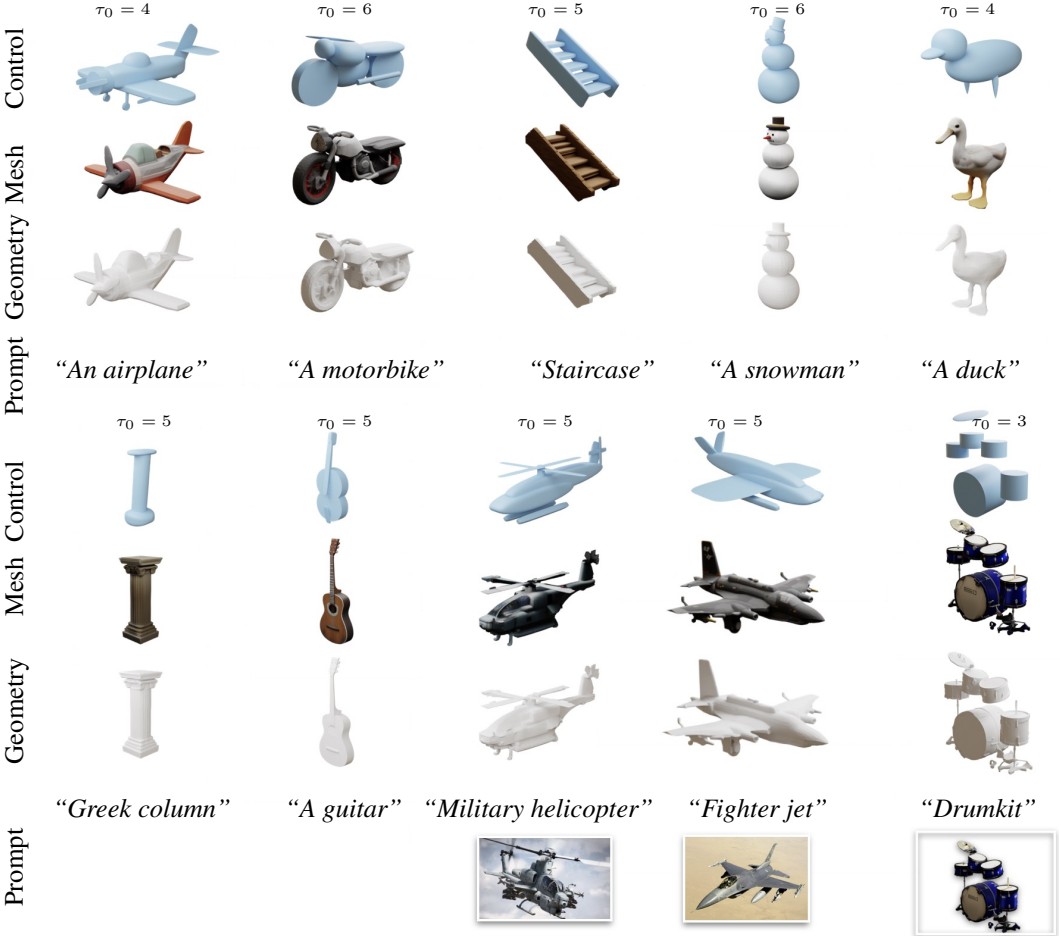

**Figure 9: Coarse and fine-grained control with superquadrics.** Superquadrics offer both fine-grained spatial control when used to sculpt precise geometry (*motorbike, staircase, helicopter*) and coarse control, when only used to draft a 3D sketch (*duck, drumkit*).

## A.3 FINE-GRAINED ALIGNMENT WITH STATE-OF-THE-ART METHODS

In Fig. 10 we show the results for the same experiment provided in the main paper, but with different control strengths.

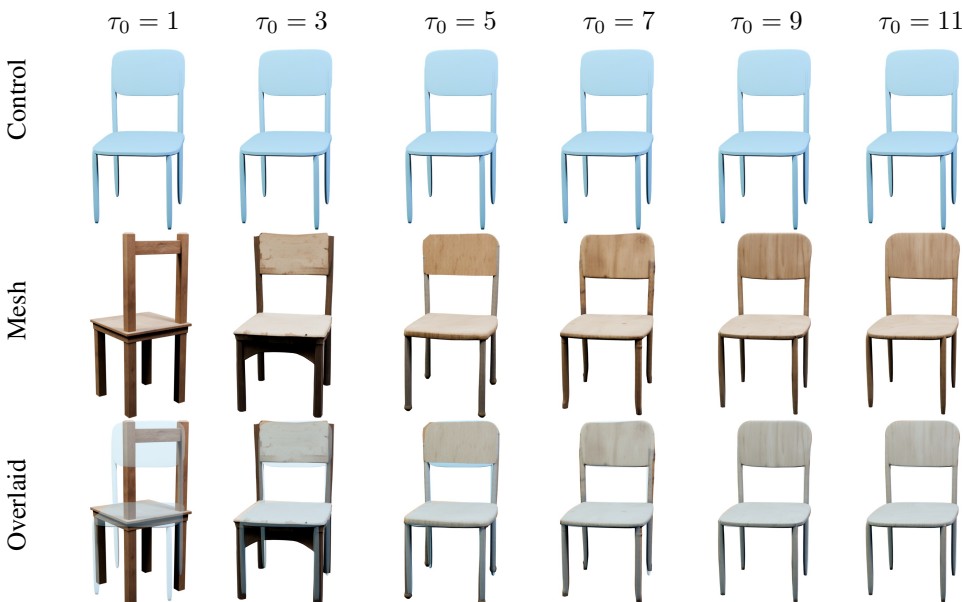

**Figure 10: Fine-grained alignment of SPACECONTROL with different $\tau_0$.** In the first row we show the input control, in the second the generated asset and in the third, we overlay the two, to better visualize alignment. All the generations use the same spatial control and the same prompt *"A wooden chair."*.

Furthermore, in Fig. 11 we show a practical application when fine-grained spatial control can be particularly useful. With our method, a user can provide a sketch of the geometric primitives composing the scene and directly condition the generation on this input, without requiring any time-consuming post-processing to align the generated shapes.

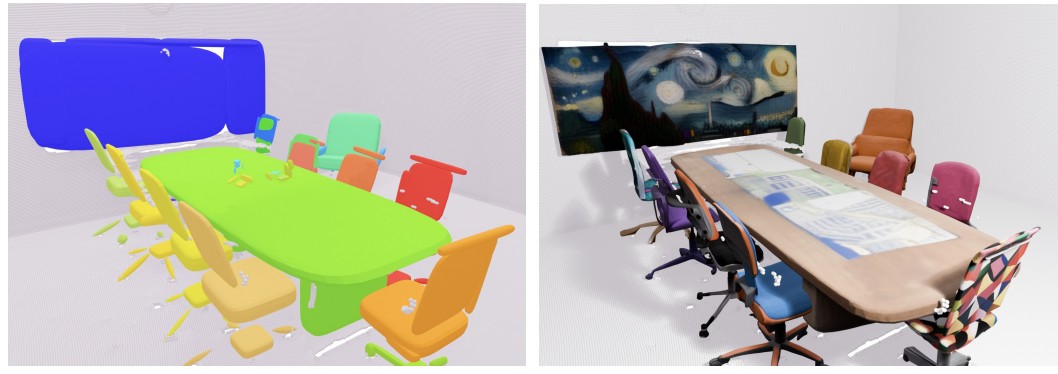

Input spatial control          Output 3D assets

**Figure 11: SPACECONTROL for 3D scene generation.** We show how SPACECONTROL can be used to generate objects of full scenes starting from a coarse conditioning. On the left we show the superquadrics for the scene, where each object is represented with a different color. On the right we show the assets generated with SPACECONTROL using the geometric primitives from the right as spatial condition. Note that each object is generated independently, by scaling the superquadrics to unit cube and giving them as spatial control to SPACECONTROL. Generated objects are then automatically placed, by undoing the transformation.

## A.4 LOCAL CONTROL

Explicit 3D geometric conditioning also paves the way for *local semantic conditioning*. While a full exploration goes beyond the scope of this work, we implemented a baseline approach which demonstrates that part-level semantic control can be achieved without any conceptual modification, providing a solid foundation for future work.

In our setting, shape conditioning already determines the geometry of the generated object . Therefore, we are interested in using *local semantic conditioning* to control the *visual appearance* of the final generation. We consider the case where the input geometric conditioning is given as superquadrics and the user defines both a global semantic prompt which defines the global semantic of the object and some local semantic prompts, which attached to each superquadric and define their respective semantics. We use the geometry of the conditioning superquadrics together with the global prompt to generate the geometry of the object in the Structure FM. Once obtained the geometry of the object, the local semantic prompts are then used to condition the visual appearance generation, in the Appearance FM. In order to combine the local and the global conditioning, we modify the cross-attention layer of the Appearance FM DiTs so that instead of only cross-attending to a global prompt, points also cross-attend to the corresponding local prompts. In practice, we first cross-attend separately to each conditioning prompt, and the feature of a point $i$ is updated with the linear combination of the cross-attention performed with *global* ($c_{global,i}$) and *local* ($c_{local,i}$) conditioning:

$$z_i \leftarrow 0.5 \cdot \text{CA}(z, c_{global,i}) + 0.5 \cdot \text{CA}(z, c_{local,i}) \, ,$$

where $c_{local,i}$ is the local semantic conditioning of the nearest superquadric to point $i$. A qualitative result of this approach in Fig. 12, where we show how this approach can be used to generate a white chair with a red seat.

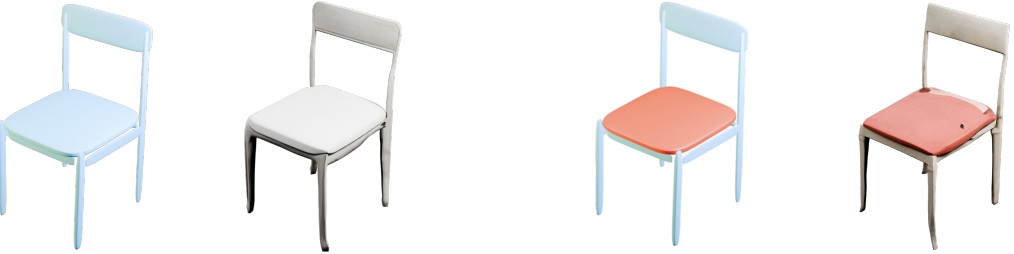

**Without** *local* semantic conditioning.       **With** *local* semantic conditioning.

**Figure 12: Local semantic control.** From left to right we show: the input geometric control, the 3D asset generated by globally conditioning on "*A white chair.*", the 3D asset generated by conditioning globally on "*A white chair.*" and locally (on the superquadric highlighted in red) on "*A read seat.*".

## B   INTERACTIVE USER INTERFACE

In Fig. 13 we visualize our interactive user interface. Starting from scratch or from a template of superquadrics, users can freely edit superquadrics using their parameters, and add/delete them. Once given the conditioning, they can select a control strength (higher control strength means that the generated shape looks more like the primitives) and a text (and optionally image) conditioning. They can then toggle between the input primitives and meshes and proceed with new generations. We provide a demo of the user interface in the supplementary video.

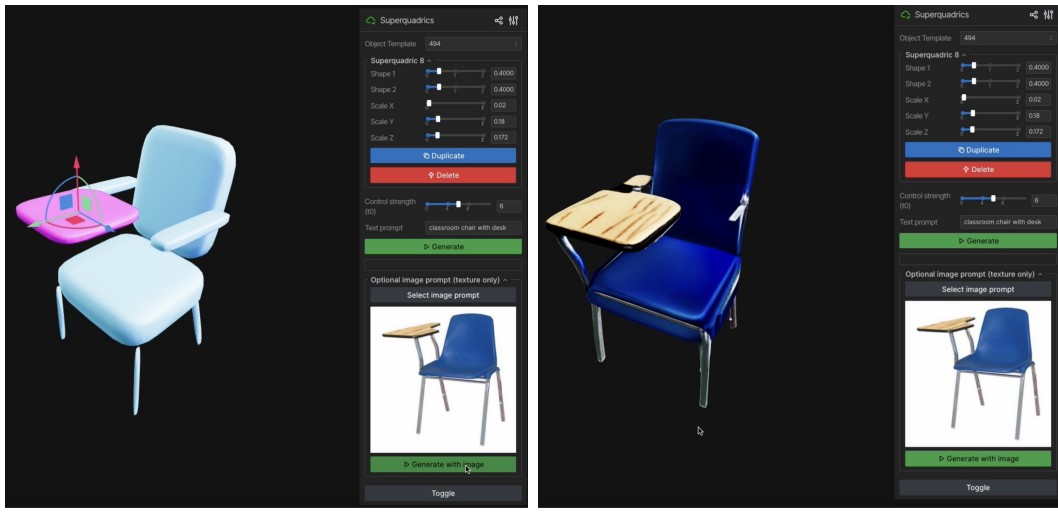

**Figure 13: Visualization of our interactive user interface.** Users can control the generated geometry by changing the shape of the geometric primitives and deciding the strength of the conditioning. Other than spatial control, users can use text and, optionally, images.

## C   USER STUDY

In Fig. 14, we show the web interface of our user study. From left to right, we show the given control shape, and two competing methods. The participants then choose which generated object is more faithful to the input control shape, which model looks more realistic, and which one they like best.

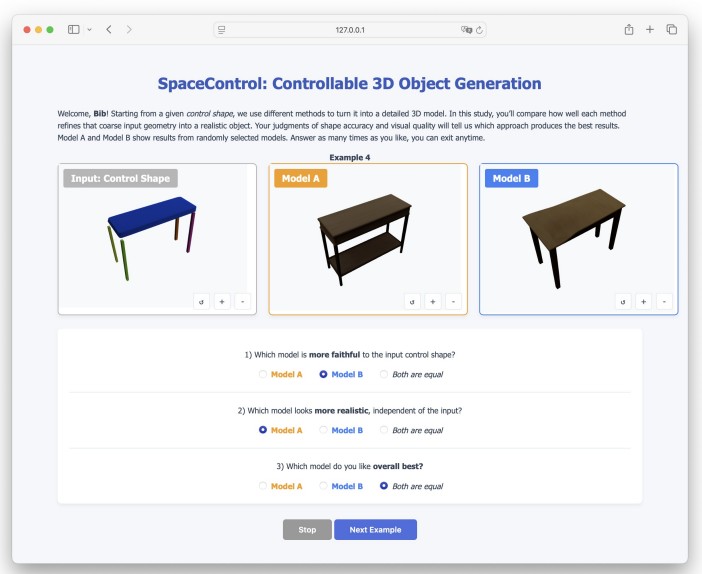

**Figure 14: User study interface.**

## D   SPICE-E-T

We obtain our training-based baseline SPICE-E-T by adding an additional conditioning layer to the flow transformer blocks in the structure generator of text-conditioned Trellis model (see Fig. 15) which perform cross attention on the shape conditioning. We encode the shape conditioning using the Trellis encoder $\mathcal{E}$, and we perform the Cross-Attention in that feature space. We initialize the original layers with the weights from the text-conditioned Trellis and the newly added ones randomly. We then train the modified *Structure Generator* for 120.000 iterations with a batch size of 4 on the ABO dataset (Collins et al., 2022), where the shape conditioning are obtained by running SuperDec (Fedele et al., 2025). During training, we use the same reconstruction loss of the original Trellis model.

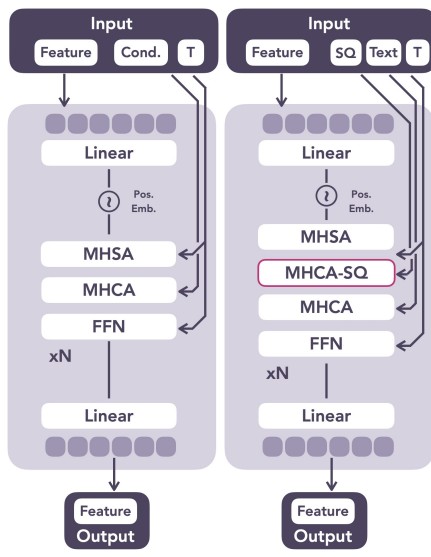

**Figure 15:** Comparison between the Flow Transformer from the original Trellis *(left)* and the one from SPICE-E-T *(right)*, adapted to enable spatial control via superquadrics.

