# OpenReview forum: "SpaceControl: Introducing Test-Time Spatial Control to 3D Generative Modeling"
_ICLR.cc/2026/Conference — ICLR 2026 Poster_

### Official Review · Reviewer_Lb61 · 2025-10-14

**Soundness:** 2
**Presentation:** 2
**Contribution:** 2
**Rating:** 4
**Confidence:** 4

**Summary:**

SpaceControl is a training-free method that introduces explicit spatial control into 3D asset generation by conditioning a pre-trained generative model (Trellis) on user-provided 3D geometry, such as superquadrics or meshes. Unlike existing approaches that require fine-tuning or optimization, SpaceControl injects geometric guidance directly into the latent space during inference using a method inspired by SDEdit. This allows users to intuitively control the shape of generated assets while preserving the realism and generalization of the base model. The method supports multi-modal inputs (text, image, and geometry) and offers a tunable parameter to balance faithfulness to the input geometry with output realism. Extensive experiments and a user study show that SpaceControl outperforms both training-based and guidance-based baselines in terms of geometric alignment and visual quality.

**Strengths:**

1. Novel Training-Free Pipeline for 3D Spatial Control
SpaceControl introduces a new inference-time guidance mechanism that enables spatial control without any fine-tuning. This is a significant departure from existing methods like Spice-E, which require task-specific training, or optimization-based methods like Coin3D, which are slow and computationally heavy.


Generality and Flexibility: Because it does not modify the model weights, SpaceControl can be applied to any pre-trained Trellis model and supports a wide range of geometric inputs—from simple primitives to detailed meshes—without retraining.

2. Tunable Faithfulness–Realism Trade-off
A key contribution is the introduction of the control strength parameter τ₀, which allows users to smoothly interpolate between:

High Faithfulness: When τ₀ is large, the output closely adheres to the input geometry.

High Realism: When τ₀ is small, the output aligns more with the data distribution of the base model, often resulting in more natural-looking assets.

3. Multi-Modal Conditioning and Practical Usability
SpaceControl supports text, image, and geometry conditioning in a unified framework: Text guides the semantic content. Image influences appearance and style. Geometry controls the 3D structure.

**Weaknesses:**

1. Limited Methodological Innovation Relative to Spice-E

While the authors position SpaceControl as a novel training-free alternative, the core idea is conceptually similar to Spice-E, which also conditions generation on geometric primitives. The main difference lies in the implementation: Spice-E uses fine-tuning, while SpaceControl uses inference-time guidance.

Heavy Reliance on Trellis: The method is built directly on Trellis and does not introduce a new generative framework. Its success is largely dependent on the capabilities of the underlying model, which limits the perceived novelty.

2. Narrow Experimental Scope and Limited Category Diversity

The experiments are conducted on a limited set of object categories—primarily chairs and tables from ShapeNet, and toys from Toys4K. This raises questions about the method’s scalability and generalization to more complex or diverse shapes.

Lack of Complex Categories: There is no evaluation on categories with intricate geometry or high structural variability (e.g., humans, animals, vehicles, or scene-level generation).

3. Manual Hyperparameter Tuning and Global Control
The control strength τ₀ is a global, manually tuned parameter that applies uniformly to the entire object. This can be a limitation in practice:

No Part-Level Control: Users cannot specify that certain parts of the object should strictly follow the geometry while others can vary freely. This limits fine-grained creative control.

Per-Instance Tuning Required: The optimal τ₀ may vary across objects, requiring users to manually experiment for each generation, which hinders fully automated pipelines.

4. Limited Comparison to Other Training-Free 3D Methods

The paper compares only to Coin3D among training-free methods. Other relevant inference-time 3D guidance approaches—such as those based on score distillation sampling (SDS) or prompt-based editing in 3D—are not discussed or evaluated, leaving the reader uncertain about how SpaceControl fits into the broader landscape of training-free 3D control.

**Questions:**

1. Limited Methodological Innovation Relative to Spice-E
2. Narrow Experimental Scope and Limited Category Diversity
3. Manual Hyperparameter Tuning and Global Control
4. Limited Comparison to Other Training-Free 3D Methods

---

> ### Author Response · Authors · 2025-11-26
>
> We thank the reviewer for the constructive and detailed feedback. In particular, we appreciate the recognition of **“generality and flexibility”**, **“tunable faithfulness-realism trade-off”**, **“multi-modal conditioning and practical usability”**. We also appreciate that our 3D guidance-based method is recognized as “a significant departure from existing methods”. In the following lines, we address the remaining concerns.
>
> ---
>
> **1. Innovation compared to Spice-E**
> The reviewer sees our method as *"a significant departure from existing methods as Spice-E"*, but is at the same time concerned about the *"conceptual similarities to Spice-E"* -- we agree that the broad *task* of conditioned 3D generation is also explored in Spice-E, but our method is *methodologically fundamentally different* from it (as the reviewer itself highlighted), as Spice-E needs *task-specific fine-tuning* of additional cross-attention layers, while our approach is *completely training free*.
>
> ---
>
> **2. Reliance on Trellis**
> The reviewer suggests that our approach is closely tied to Trellis, and does not introduce a *novel generative framework*. In fact, the focus of this work is not to introduce a novel generative framework, but a *novel way to control* existing generative models, such as Trellis. In practice, our method is generally applicable to any model that leverages diffusion or flow-matching for generating the 3D object geometry. This paradigm is used by most recent top-performing models, including the very recently released SAM-3D.
>
> ---
>
> **3. Reliance on manual parameter tuning**
> The reviewer asks whether the choice of the control parameter $\tau_0$ can be automated to allow SpaceControl to be used in fully automated pipelines. That is correct: in principle, $\tau_0$ is intended to give users a **flexible** control of the level of adherence to the geometric prompt. However, when the generation needs to be automated, $\tau_0$ can be dynamically adapted to meet desired geometric and perceptual constraints, which can be evaluated with *automated metrics* (e.g. Chamfer Distance and FID), without any need of human intervention.
> We will clarify this point in the revised manuscript.
>
> ---
>
> **4. Part-Based Control**
> The reviewer notes that our primitive-based conditioning could, in principle, be extended to support part-level semantic control, and asks whether SpaceControl can accommodate such localized modifications. We thank the reviewer for this suggestion, and we agree that part-based control is fully compatible with the design of SpaceControl. We implemented this idea in which each geometric primitive (when using superquadrics as conditioning) receives a local semantic prompt in addition to the global one.
> This minimal modification enables meaningful localized edits: as shown in **Fig. 7 of the updated manuscript** a chair generated with the global prompt “white chair” can have only its seat modified using the local prompt “red seat” while the rest of the geometry remains unchanged. This demonstrates that part-level semantic or geometric control can be achieved without any conceptual modification, and provides a solid foundation for future work.
>
> ---
>
> **5. Category Diversity of Evaluation**
> The reviewer is concerned about the object diversity in the evaluations datasets, and specifically suggests evaluating on object classes such as **animals** or **vehicles**. In fact, animals and vehicles are part of Toys4K together with plants, and other everyday objects. We strictly follow the same evaluation setup as Trellis, and additionally evaluate on ShapeNet for a fair comparison with Spice-E. Throughout the paper we additionally show qualitative results on many other object types.
>
> ---
>
> **6. Comparison to More Training-Free 3D Methods**
> The reviewer asked to compare with other training-free approaches, but did not provide any recommendation. We did not find any other relevant approach which solves the task of fine-grained spatial control for 3D asset generation. We are happy to compare with additional methods provided by the reviewer.

---

### Official Review · Reviewer_VKm8 · 2025-10-31

**Soundness:** 3
**Presentation:** 3
**Contribution:** 3
**Rating:** 4
**Confidence:** 4

**Summary:**

SpaceControl is a training-free, test-time method that introduces explicit spatial control into the pre-trained 3D generative model Trellis. It achieves precise geometric control by encoding user-provided geometric inputs—such as superquadrics or meshes—into the model’s latent space and guiding the denoising process accordingly. The method supports adjustable control strength via a parameter τ₀, enabling users to trade off between geometric fidelity and visual realism. It outperforms both training-based and optimization-based baselines across multiple datasets and includes an interactive interface for real-time editing and generation.

**Strengths:**

1. The motivation is clear and practical. Allowing users to directly manipulate geometry using 3D primitives, aligns well with real-world design workflows.
2. The paper is well-structured, with a logical flow and clear figures.
3. The performance is good and includes a user study and interactive interface to demonstrate practical utility.

**Weaknesses:**

1. Limited technical novelty. The core idea of injecting geometric guidance before the appearance generation stage appears to be a natural extension of Trellis’s inherent two-stage design (coarse geometry ->  fine appearance). This raises the question of whether the controllability stems more from Trellis itself than from a genuinely novel contribution.
2. Potential generalizability problem beyond two-stage architectures. The method is tightly coupled to Trellis’s specific disentangled geometry-appearance pipeline. It is unclear how SPACECONTROL would adapt to end-to-end 3D generative models or single-stage frameworks, limiting its broader applicability.
3. Inadequate baselines. The comparison primarily includes methods published before or during 2024, missing recent advances in controllable 3D generation (e.g., 2025 works cited in the introduction). This weakens the claim of state-of-the-art performance and raises concerns about the completeness of the evaluation.
4. Missing References. The idea of 3D-reference-based 3D generation is similar to ThemeStation [siggraph 2024] and Phidias [iclr 2025], but only coin3d is compared.

**Questions:**

1. Generalizability to other 3D generative models: The method is tightly integrated with Trellis’s two-stage geometry–appearance pipeline. Can SPACECONTROL be adapted to other 3D generative architectures? and if so, how does its performance compare in terms of faithfulness and realism?
2. Extension to richer conditioning modalities: Currently, spatial control relies on 3D primitives (e.g., superquadrics) combined with text or optional image prompts. Could the framework naturally incorporate additional conditioning signals—such as depth maps, sketches, or multi-view images—to enable more expressive and intuitive user control?
3. Necessity and timing of geometric guidance: The geometric prior is injected only at the start of the structure generation stage (i.e., after initial noise). Given that Trellis already disentangles geometry and appearance, why not introduce spatial guidance even earlier (e.g., at ) or jointly optimize both stages with geometric constraints? Is the current design a limitation of the rectified flow formulation or a deliberate trade-off?

---

> ### Author Response · Authors · 2025-11-26
>
> We thank the reviewer for their constructive and detailed feedback. In particular, we appreciate that they recognize our task as **“clear and practical”**, **“well aligned with real-world design workflows”** and of **“practical utility”**. Below, we address their remaining questions.
>
> ---
>
> **1. Generalizability to Other Generative Models**
> The reviewer is curious whether our approach is limited to Trellis. This is not the case, in fact our method is generally applicable to any model that leverages diffusion or flow-matching for generating the 3D object geometry. This paradigm is used by most recent top-performing models, including the newly released SAM-3D, so it does not pose a practical limitation within the current landscape of generative 3D models.
>
> ---
>
> **2. Additional Input Modalities**
> The reviewer asks whether our approach extends beyond superquadrics. In the paper we demonstrate results with both superquadrics and meshes, and also condition on text and image inputs.
> The reviewer suggests *depth maps*, *sketches*, and *multi-view images*:
> - Depth maps can be used directly, as they can be provided to the model in the same manner as meshes.
> - Sketches are conceptually aligned with our use of superquadrics, which can be seen as a lightweight 3D sketch interface, which is what we target in this work.
> - TRELLIS natively already supports multi-view images which our method inherits, though we would note that they are not necessarily more intuitive than superquadrics. Images are difficult to modify interactively without additional generative models, and producing multi-view-consistent images is even less trivial.
>
> We will include these clarifications in the main paper.
>
> ---
>
> **3. Other Baselines**
> The reviewer notes the absence of comparisons to more recent 2025 methods mentioned in our introduction. We would like to clarify that our baselines focus on the most recent approaches targeting *spatially controllable* 3D object generation, which is the specific task of this work. The cited 2025 methods Hunyuan3D-2.0 and 3DTopia-XL are designed for *text- or image-conditioned* 3D generation, while Instant3DIt is a mesh-editing method that requires an input mesh and mask. None of these are directly applicable to *spatially controllable* generation, and thus there is no straightforward comparison setting.
>
> ---
>
> **4. Timing of Guidance**
> The reviewer asks whether spatial guidance could be applied even *before* the initial noise sampling. Since noise initialization is the first step in diffusion/flow-matching models, it is unclear how conditioning could be injected at an earlier point. Our method begins from a latent encoding of the geometric guidance, applies noise, and then denoises following Trellis trajectories under semantic control (coming from text/images); we therefore do not see a natural mechanism for conditioning prior to this. We would be happy to expand on this point if clarification is needed.
> The reviewer also asks why spatial guidance is not applied in both stages of Trellis. The first stage is responsible for the geometry, so conditioning here directly influences structure. The second stage refines only texture and fine-grained structure details, without modifying the overall geometry, so spatial constraints introduced there would not affect shape.
>
> ---
>
> **5. Missing References**
> The reviewer highlights *ThemeStation* and *Phidias*, which focus on theme- and reference-guided object generation. While these works target different problems, they are related in spirit, and we will incorporate a discussion of their connection into the updated related work section.

---

### Official Review · Reviewer_W86Y · 2025-11-01

**Soundness:** 4
**Presentation:** 3
**Contribution:** 3
**Rating:** 6
**Confidence:** 4

**Summary:**

The paper proposes a method to add explicit spatial conditioning to a pretrained 3D generative model (Trellis) at test time, without retraining. Users provide geometric inputs, coarse superquadrics, which enables explicit control.

It performs latent-space interpolation between the control geometry’s embedding and random noise, then uses the pretrained structure-flow and appearance-flow models to denoise and decode the final 3D asset. A scalar parameter governs the trade-off between faithfulness to the spatial control and visual realism.

Experiments demonstrate that this training-free strategy yields stronger geometric fidelity than baselines while maintaining comparable realism scores (slightly reduced). The work also presents a browser-based interface for interactive shape manipulation through superquadrics.

**Strengths:**

**Training-free controllability.**
The whole system is designed without model training or surgery, which preserves the strong 3D generative prior in Trellis [1], and the approach introduced seems to be also applicable to other / future 3D generation under the Trellis framework.

**Good controllability.**
With superquadrics, this method enables explicit control with only a few primitives, which is user-friendly and accurate on most shapes. It also introduces a tunable adherence parameter that enables test-time adaptation according to actual usage.

**Empirical quality.**
This method consistently improves geometric alignment relative to strong baselines and Trellis itself.

```
[1] Trellis: Structured 3D Latents for Scalable and Versatile 3D Generation.
```

**Weaknesses:**

**Implied control limitation on curvy shapes.**
The 3D geometry representation, superquadrics, is often suitable for objects primarily containing convex shapes - cubes, spheres, cylinders, etc., while for concave shapes, it requires multiple superquadrics to compose. In pure 3D reconstruction domain, this does not impose a problem, but in terms of controllability, where simplicity or the number of the underlying primitives is more critical, superquadrics may suffer.

**Limited conceptual novelty.**
The latent-interpolation mechanism is directly analogous to image-domain test-time editing (e.g., SDEdit [1]) and does not introduce fundamentally new generative principles.

**Architecture dependency.**
The approach assumes access to a pretrained model with paired latent encoder and flow decoder (Trellis [2]), which means its generality to other 3D generative architectures is uncertain.

**Realism and appearance degradation.**
Geometrc metrics (Chamfer distance) is improved greatly, but this seems to be at the expense of realism (CLIP-I, FID) when compared to Trellis in Tab. 1. This means the proposed method is still a trade-off, just like the previous methods.


```
[1] SDEdit: Guided Image Synthesis and Editing with Stochastic Differential Equations.
[2] Trellis: Structured 3D Latents for Scalable and Versatile 3D Generation.
```

**Questions:**

1. How many superquadircs would it need to precisely represent the elephant trunk shown in Fig. 3? I understand the current is one and rely on sacrificing controllability to generate this curve trunk, but what would the approximate amount of superquadrics if we want an accurate control?

---

> ### Author Response · Authors · 2025-11-26
>
> We thank the reviewer for the constructive and detailed feedback. In particular, we are happy to see that our training-free approach is appreciated for its controllability and for the flexibility to be **“applicable to future 3D generation under the Trellis framework”**. The reviewer also appreciates the **”empirical quality”** of our method, which **“consistently improves geometric alignment relative to strong baselines and Trellis itself”**. Finally, we are happy to see that the reviewer appreciates our browser-based user interface. Below, we address the remaining questions and concerns.
>
> ---
>
> **1. Control with Curvy Shapes**
> The reviewer asks whether curved shapes can serve as spatial control signals, noting that superquadrics primarily model convex forms. This is correct, yet superquadrics admit well-established extensions for curving and tapering, enabling concave geometries [1, 2]. We added an illustration of these curved/tapered primitives in the revised manuscript (Fig. 9). Importantly, SpaceControl is not restricted to superquadrics, any 3D representation, from coarse primitives to high-resolution meshes, can be used as conditioning.
>
> [1] Jaklic, Leonardis, Solina “Segmentation and Recovery of Superquadrics”. [Springer Science 2000]
> [2] Pelossof et al. “An SVM Learning Approach to Robotic Grasping.” [ICRA 2004]
>
> ---
>
> **2. The Elephant's Trunk**
> The reviewer asks how many superquadrics would be required to model the elephant trunk in Fig. 3. In that example we deliberately use a single primitive to highlight robustness under coarse spatial guidance; if finer control of curvature is needed, users may specify several superquadrics or adopt more expressive primitives. SpaceControl readily accommodates such geometric conditioning. To illustrate this, the supplementary video includes an example of independently modeling humans fingers using multiple superquadrics.
>
> ---
>
> **3. Realism and Appearance Degradation**
> We agree that very large values of the control strength parameter $\tau_0$, combined with very coarse geometric conditioning, can pull the sample away from Trellis’s training manifold and lead to a drop in visual quality. However, this effect does not occur in the majority of cases. As shown in Table 2, when the conditioning geometry is *more detailed* (M columns), the FID scores for high $\tau_0$ values are in fact *the best across all the baselines*, indicating that conditioning on detailed meshes can even improve realism relative to the base model. When the conditioning is given as *coarse geometric primitives* (P columns), FID only degrades for *extreme* $\tau_0$ values, which we do not recommend in practice. Therefore, in the typical operational range of $\tau_0$, SpaceControl does not sacrifice high visual quality while providing substantially improved geometric fidelity. We will clarify this aspect in the revised manuscript.
>
> ---
>
> **4. Architecture Dependency**
> The reviewer asks whether our approach is limited to Trellis. This is not the case, in fact our method is generally applicable to any model that leverages diffusion or flow-matching for generating the 3D object geometry. This paradigm is used by most recent top-performing models, including the newly released SAM-3D, so it does not pose a practical limitation within the current landscape of generative 3D models.

---

### Official Review · Reviewer_wmdV · 2025-11-03

**Soundness:** 3
**Presentation:** 3
**Contribution:** 3
**Rating:** 6
**Confidence:** 4

**Summary:**

This paper introduces SPACECONTROL, a novel training-free, test-time method for adding explicit spatial control to pre-trained 3D generative models. Addressing the limitations of ambiguous text prompts and cumbersome image editing, SPACECONTROL enables users to guide 3D asset generation using diverse geometric inputs, including primitives and meshes. The method works by encoding the user-specified geometry into the latent space of a powerful pre-trained model and using this latent code to initiate the denoising process from an intermediate timestep. A key contribution is a controllable parameter that allows users to fluidly trade off between geometric faithfulness and output realism. Extensive quantitative evaluations and user studies demonstrate that the method outperforms existing training-based and optimization-based baselines in geometric fidelity while preserving high visual quality. The authors also present an interactive user interface for practical creative workflows8888.

**Strengths:**

1: The proposed SPACECONTROL is a training-free method that injects explicit spatial control into a powerful pre-trained 3D generative model (Trellis) purely at test-time, which is efficient.

2: The framework provides intuitive geometric control using diverse inputs, from coarse primitives to detailed meshes, and introduces a single parameter that allows users to flexibly trade off between geometric faithfulness and output realism.

3: Extensive experiments show the method significantly outperforms both training-based (Spice-E, SPICE-E-T) and guidance-based (Coin3D) baselines in geometric alignment (measured by Chamfer Distance) while preserving high visual quality (measured by FID and P-FID).

4: This paper also presents a practical interactive user interface that enables online editing of superquadrics for their real-time conversion into detailed, textured 3D assets, demonstrating its high utility for creative workflows.

**Weaknesses:**

1: **Reliance on Manual Parameter Tuning.** The crucial trade-off between realism and faithfulness is governed by the $\tau_0$ parameter, which the paper states must be "selected manually"2. While Table 2 and Figure 3 show the effect of varying $\tau_0$, the optimal value appears to be object-dependent. This manual, per-instance tuning requirement undermines the method's practicality and hinders its use in automated generation pipelines.

2: **Lack of Part-Based Control.** The parameter $\tau_0$ is applied globally, "enforcing a uniform adherence level across the entire object"4. This is a significant limitation, as it prevents more nuanced, part-aware editing. For example, a user cannot simultaneously enforce high geometric fidelity for one part (e.g., the legs of a chair) while allowing high realism and creative variation for another (e.g., the backrest).

3: **Potential Information Bottleneck from Voxelization.** The method requires all spatial control inputs, including "detailed meshes," to be first voxelized into a $64 \times 64 \times 64$ grid before being passed to the encoder. This coarse, low-resolution representation may act as an information bottleneck, losing the very "fine-grained" details the user wishes to control. The paper does not analyze the impact of this $64^3$ resolution limit on the final geometric faithfulness.

4: **Limited Analysis of Failure Cases.** The paper presents many successful qualitative results (e.g., Figures 1, 4, 7, 8) but lacks a dedicated analysis of failure cases. For instance, what happens when the text prompt and spatial control are semantically contradictory (e.g., text="a car", spatial control=a boat)? Or when the control geometry is highly complex or topologically different from the objects in the pre-trained model's prior?

**Questions:**

1: **Auto optimization of $\tau_0$.** The paper states that the adherence parameter $\tau_0$ must be "selected manually" and that this is a limitation. Could the authors elaborate on the challenges of automating this selection? Have they explored any methods to predict an optimal or suggested $\tau_0$ based on the properties of the input control geometry (e.g., its complexity) or its relation to the text prompt? Clarification on whether this is a fundamental, subjective trade-off or a solvable engineering problem would be valuable.

---

> ### Author Response · Authors · 2025-11-26
>
> We thank the reviewer for the constructive and detailed feedback. In particular, we appreciate that our geometric conditioning framework is perceived as **“efficient”** as it is training-free, and **”flexible”** as it relies on a single control parameter, which is a **“key contribution”** and allows users to flexibly decide the strength of the conditioning. The reviewer also appreciates our **”extensive experiments”** and our **“practical interactive user interface”**. Below, we address the remaining questions and concerns:
>
> ---
>
> **1. Automatic optimization of $\tau_0$**
> The reviewer asks whether the choice of the control parameter $\tau_0$ can be automated to allow SpaceControl to be used in fully automated pipelines. That is correct: in principle, $\tau_0$ is intended to give users a *flexible* control of the level of adherence to the geometric prompt. When the generation needs to be automated, $\tau_0$ can be dynamically adapted to meet desired geometric and perceptual constraints, which can be evaluated with automated metrics (e.g. Chamfer Distance and FID).
> We will clarify this point in the revised manuscript.
>
> ---
>
> **2.Part-Based Control**
> The reviewer notes that our primitive-based conditioning could, in principle, be extended to support part-level semantic control, and asks whether SpaceControl can accommodate such localized modifications. We thank the reviewer for this suggestion, and we agree that part-based control is fully compatible with the design of SpaceControl. We implemented this idea in which each geometric primitive (when using superquadrics as conditioning) receives a local semantic prompt in addition to the global one.
> This minimal modification enables meaningful localized edits: as shown in **Fig. 8 of the updated manuscript** a chair generated with the global prompt “white chair” can have only its seat modified using the local prompt “red seat” while the rest of the geometry remains unchanged. This demonstrates that part-level semantic or geometric control can be achieved without any conceptual modification, and provides a solid foundation for future work.
>
> ---
>
> **3. Voxelization**
> The reviewer asks whether voxelizing the geometric conditioning before passing it to the encoder introduces an information bottleneck that prevents faithful conditioning on fine-grained details. We agree that voxelizing the input geometry may, in principle, discard some details. However, our conditioning mechanism uses the *native $64^3$ resolution* of Trellis’s structure encoder, meaning that SpaceControl does not introduce any additional bottleneck beyond what Trellis already relies on during training. Empirically, this resolution is sufficient for detailed conditioning: as shown in Fig. 9 and 10 of the supplementary material, fine structural details provided through primitive-based guidance are preserved in the final generation despite voxelization. We will add a clarification to avoid the impression that voxelization inherently limits fidelity.
>
> ---
>
> **4. Semantically Contradictory Conditioning**
> The reviewer asks how SpaceControl behaves when the geometric control and the textual prompt are semantically inconsistent, e.g., prompting “a car” while providing the geometry of a boat. Handling adversarial combinations of geometry and text is not the target of our work, but we tested SpaceControl in this challenging scenario. As suggested by the reviewer, we sketched a boat using superquadrics and prompted the model twice: once with “a boat” and once with “a car” (see Fig. 7 of the revised document). As expected, the “boat” prompt yields a coherent result that reflects both the geometry and the text. With the “car” prompt, the model still adheres to the boat geometry but injects car-like appearance cues, producing an asset that visually resembles a car while retaining the overall boat structure. Even though SpaceControl naturally performs best when text and geometry are aligned (i.e., when the sample remains near Trellis’s training distribution), this experiment shows that the method can still generate reasonable outputs even under contrasting prompts.

---

### Comment · Area_Chair_fFV3 · 2025-11-27

Dear Reviewers,

Thank you for your efforts in evaluating this submission. The current set of reviews shows a divergence in the overall scores. To ensure a fair and well-informed final decision, it is important that we have active participation from all reviewers during the author-reviewer discussion phase.

The authors have now responded to your comments. I kindly ask each of you to review their replies and engage in the discussion, especially to clarify whether their responses address your concerns and whether your initial assessment remains the same.

Your contributions at this stage are crucial for reaching a balanced consensus.
Thank you again for your time and commitment to the review process.

Best regards,

Area Chair

---

### Meta-Review · Area_Chair_rF6k · 2026-01-06

**Summary:**

The paper received a split vote (scores: 6, 6, 4, 4), placing it on the borderline. The primary debate centered on the tradeoff between the method's practical utility (training-free, efficient) and its perceived technical novelty (similarity to SDEdit).

**Reviewer Concerns:**

The authors provided a high-quality rebuttal. Crucially, they addressed the critique regarding "lack of part-based control" (raised by wmdV and Lb61) by implementing localized semantic editing. They also clarified why recent 2025 baselines suggested by VKm8 were not comparable and explained the automation of the control parameter $\tau_0$. The concern regarding limited conceptual novelty (Reviewer W86Y noting the similarity to SDEdit) remains valid but is offset by the significant practical utility of enabling spatial control  without the heavy computational cost of retraining or fine-tuning.

**Reviewer Scores:**

The rebuttal strengthened the paper by removing a functional limitation. Reviewers wmdV (Score 6) and Lb61 (Score 4), who specifically asked for part-based control, would likely have raised their scores had they engaged. Reviewer VKm8's concerns regarding baselines were factually addressed. Reviewer W86Y was already supportive. The consensus is that the method's efficiency and flexibility outweigh the incremental nature of the underlying mechanism.

---

### Decision · Program_Chairs · 2026-01-26

Accept (Poster)